# Nanopore Sequencing Assessment of Bacterial Pathogens and Associated Antibiotic Resistance Genes in Environmental Samples

**DOI:** 10.3390/microorganisms11122834

**Published:** 2023-11-22

**Authors:** Andrei Lobiuc, Naomi-Eunicia Pavăl, Mihai Dimian, Mihai Covașă

**Affiliations:** 1Department of Biomedical Sciences, Faculty of Medicine and Biological Sciences, “Ştefan cel Mare” University, 720229 Suceava, Romania; naomi.paval@usm.ro (N.-E.P.); mcovasa@usm.ro (M.C.); 2Department of Computers, Electronics and Automation, Stefan cel Mare University of Suceava, 720229 Suceava, Romania; dimian@usm.ro

**Keywords:** Oxford Nanopore Technology, wastewater, microbes, 16S, tetracycline, treatment, meat

## Abstract

As seen in earlier and present pandemics, monitoring pathogens in the environment can offer multiple insights on their spread, evolution, and even future outbreaks. The present paper assesses the opportunity to detect microbial pathogens and associated antibiotic resistance genes, in relation to specific pathogen sources, by using nanopore sequencing in municipal waters and wastewaters in Romania. The main results indicated that waters collecting effluents from a meat processing facility exhibit altered communities’ diversity and abundance, with reduced values (101–108 and 0.86–0.91) of Chao1 and, respectively, Simpson diversity indices and *Campylobacterales* as main order, compared with other types of municipal waters where the same diversity index had much higher values of 172–214 and 0.97–0.98, and *Burkholderiaceae* and *Pseudomonadaceae* were the most abundant families. Moreover, the incidence and type of antibiotic resistance genes were significantly influenced by the proximity of antibiotic sources, with either tetracycline (up to 45% of total reads) or neomycin, streptomycin and tobramycin (up to 3.8% total reads) resistance incidence being shaped by the sampling site. As such, nanopore sequencing proves to be an easy-to-use, accessible molecular technique for environmental pathogen surveillance and associated antibiotic resistance genes.

## 1. Introduction

### 1.1. Relevance of Pathogenic Microorganisms in Municipal Waters

Along with the population growth and socio-economic development, the production of domestic and industrial wastewater increases from year to year. Consequently, the biological complexity of wastewater has increased in the last few decades, and, nowadays, municipal wastewater is contaminated with a vast array of microbes, spanning all major classes of microorganisms (Figure 1) and also viruses. As such, municipal wastewater represents a potential epidemiological outbreak reservoir and a favorable environment for the evolution and dissemination of new genes and even pathogens [1].

Wastewater may have different sources, such as domestic, commercial, industrial, and agricultural sectors, that affect water quality [2]. As such, sewage water contains microbes characteristic of crop fields, animal farms, and human body inhabitants, including pathogens. The presence of such microorganisms in aquatic environments may represent a public health risk through drinking water and the consumption of food of aquatic origin [3].

The presence of bacterial pathogens in water is an important cause of many infectious diseases, such as gastroenteritis, diarrhea, melioidosis, pulmonary diseases, skin infection, lung infections, urinary tract and kidney, and others [4]. For instance, gastrointestinal infections (diarrhea, dysentery, and gastroenteritis) may appear due to exposure to bacterial pathogens, such as *Salmonella* spp., *Escherichia* spp., *Shigella* spp., *Yersinia* spp., and *V. cholerae* [2], with usual numbers for bacteria, such as *E. coli*, ranging from 2.5 × 10^3^ to 4.4 × 10^5^ colony forming units (CFU)/100 mL or total coliforms up to 7.9 × 10^5^ CFU/100 mL [5]. Another pathogen, *Clostridium perfingens*, was also detected at concentrations between 5.4 × 10^2^ and 9.1 × 10^2^ most probable number (MPN)/100 mL [5]. Other diseases caused by bacteria from water are wound infections, caused by *Pseudomonas aeruginosa*, respiratory infections, by *L. pneumophila* and *Mycobacterium avium*, and leptospirosis, caused by *Leptospira* [2].

### 1.2. Sources of Antimicrobial Resistance Genes in Municipal Waters

In addition to potentially inducing diseases, aquatic environments are suitable for the emergence or occurrence of resistance against antibiotics. In natural environments, especially aquatic ones, the presence of antibiotic resistance genes (ARGs) can have two major sources:(a)The introduction of microorganisms that have already developed resistance to antibiotics through mutational mechanisms in response to antimicrobial factors. Genes resistant to aminoglycosides and β-lactams have been identified at all stages of farming in *E. coli*, a bacterium with a habitat in the intestinal tract [6].(b)The presence in the environment of the antimicrobial agents themselves. Only 30% of the antibiotics consumed are metabolized by users (humans/animals), whilst the highest amount is released in wastewater through feces and urine or by improper disposal, thus reaching bacterial communities with high density [7].

In both cases, the sources of contamination of natural environments are represented by anthropogenic activities, such as those in the sanitary system or livestock farming. Prolonged exposure and fast multiplication times of bacteria may lead to acquiring antibiotic resistance in a few generations through several mechanisms, such as horizontal gene transfer of ARGs, genetic mutation, and recombination, favored by the existence of hypermutator bacterial strains and the proliferation of ARGs [8].

The proliferation of ARGs is favored by chemicals utilized in wastewater treatment, such as disinfectants, metals, pharmaceuticals, and other organic compounds, that influence their transmission, expression, and mobilization [9] via changing the abundance of mobile genetic elements [10].

### 1.3. Detection Methods for AMR Surveillance and Research

Monitoring pathogenic bacteria in municipal wastewater is important for managing the public health risk and for ensuring the efficiency of treatment plants. From a legislative point of view, guidelines such as those established by the US Environmental Protection Agency (USEPA 2012) [11] and the World Health Organization (WHO) [12] specify requirements for water quality monitoring. Regarding the presence of bacteria, only a few microbial indicators are used for the assessment of water quality, *E. coli* (total coliforms/fecal coliforms) being the most widely used microbial parameter in water legislation [13]. Usual methods for the detection of microorganisms in water are cultured-based, such as the membrane filtration method, presence–absence tests, direct total microbial count, and most probable number method [14].

These conventional methods for the detection of bacterial pathogens have several limitations, such as low accuracy and sensitivity, sample processing complexities, and time-consuming processes, which led to the development of new techniques. Polymerase chain reaction (PCR), enzyme-linked immunosorbent assay (ELISA), and fluorescent in situ hybridization (FISH) are several molecular methods applied in this field, but they still possess some limitations, for example, the inability to distinguish between live and dead organisms (PCR) or the low sensitivity, cross-reactivity with closely related antigens, and the need for pre-enrichment in order to reduce the cell surface antigens (immunology-based methods) [4]. Other rapid detection techniques are also utilizing biosensors, but the analysis is affected by environmental changes in pH, mass, and temperature [15,16].

Next-generation sequencing (NGS) techniques offer the capacity to accurately characterize complex microbial communities, based on genetic information, due to their large multiplexing ability. Various studies report applications of NGS technology in the characterization of bacterial communities in soil [17], drinking water [18], and pathogen identification, due to strain identification, prediction of drug resistance, and auxiliary genomic information-obtaining capacities [19]. NGS (on short reads platforms such as Illumina and Ion Torrent) allows for the analysis of hypervariable genetic regions with taxonomic significance, such as the 16S genes, with the V1–V7 regions, which are well characterized, with extensive databases [20]. On this basis, microbial communities can be easily and conveniently analyzed by initial amplification, followed by multiplex sequencing [21]. Moreover, the evolution of third-generation sequencing allows for the analysis of much longer regions, including whole genomes [22], offering greater taxonomic accuracy but also the possibility of identifying antibiotic-resistant genes [23,24]. In this sense, Oxford Nanopore Technology allows for portability and low costs for 16S or even the entire genome sequencing of the microbial community, while also offering automated bioinformatics analysis workflows. Such methods can be synergistically combined with the quantitative analysis carried out with traditional methods. However, differences in sample volume, DNA extraction method, library preparation method, NGS platform, read length, sequence depth, and data quality filtering make it difficult to offer comprehensive results in a certified system. Thus, standardized methods are needed to overcome these critical aspects of NGS [22].

Nanopore sequencing is beginning to be considered as a viable alternative for evaluating water and wastewater quality [22], enabling accurate detection to the species level of representative microbes, including pathogens of *Enterobacteriaceae*, *Bacteroides*, *Prevotella*, and other groups [25]. Due to its high-throughput nature, nanopore sequencing allows for metagenomic analyses in waters and wastewaters, with a very high degree of results overlapping with Illumina sequencing (agreed on 945 out of 999 taxons with an average abundance difference of 0.56% (median 0.15%)) [26]. Nanopore sequencing also proved its feasibility on pathogens such as those involved in meat contamination, for instance, *Campylobacter jejuni*, even allowing for comparisons with ISO standards, as described in previously research [27]. Moreover, in addition to the taxonomical identification of species, another advantage, which was validated by a side-by-side comparison of results with established sequencing platforms such as Illumina, is the identification of antibiotic resistance genes [23,24], allowing for resistome analysis in waters and wastewaters.

While microbial pathogen detection in environmental samples such as wastewaters was previously performed through sequencing, the present study aims to shed new light on the opportunity of applying a fast, affordable method to assess microbial diversity in epidemiologically relevant areas of a city. This, coupled with the bioinformatic ability to identify antibiotic resistance genes in the same samples, may serve as a novel tool for screening genetic traits that can be routinely monitored by research groups or even public health agencies to better understand the mobility of potential pathogens.

## 2. Materials and Methods

### 2.1. Sample Collection and Processing

Wastewater was sampled in August 2022 from three main areas: two streams, which collect municipal wastewater from neighborhoods of Suceava, N-E Romania, a city with approximately 84,300 inhabitants, as well as from the main river that crosses the city-Suceava river (Figure 2). The coordinates of the collection points were: S1 (47.6488, 26.2751), S2 (47.6526, 26.2782), S3 (47.6522, 26.2800), S4 (47.6550, 26.2760), S5 (47.6519, 26.2872), S6 (47.6680, 26.2779).

Samples S1 and S2 were taken from another stream (Cetatii) that collects the wastewater from the center of the city of Suceava and residual water from a pork meat processing factory. The first sampling point (S1) was in close proximity to the factory (approximately 50 m), whereas the second sample was taken from a location 200 m further downstream, before flowing into Suceava river. A third sample (S3) was collected from Suceava river, immediately after the point where Cetatii stream flows in, while S4 was collected from the same Suceava river, approximately 500 m upstream of S3. Sample S5 was collected from the exit of the city’s sewage treatment plant, and sample S6 was collected from a second stream that collects the wastewater from Burdujeni neighborhood, namely Burdujeni stream.

For each sample, triplicates of 50 mL each of water were taken into sterile tubes from beneath the surface of the water and were refrigerated until processing, and, subsequently, they were pooled in equal ratios to account for variability. The samples were centrifuged for 10 min at 4000 rpm to obtain sediment used in further analysis.

### 2.2. DNA Extraction and Quantification

DNA extraction was performed with NucleoSpin Soil kit (Macherey-Nagel, Allentown, PA, USA), according to the protocol, starting from approximately 300 mg of sediment. Cell lysis was performed with lysis buffer SL1, suitable for soil consisting predominantly of minerals. After successive washing of genetic material, the elution was carried out in 50 µL elution buffer. Samples with little or no sediment were filtered under vacuum through membrane with 0.45 µm pore size. The membrane was transferred in elution tube provided in the kit, and the lysis followed by extraction was performed according to the protocol. Quantification was performed with Qubit dsDNA Assay (Invitrogen Inc., Waltham, MA, USA) [28].

### 2.3. Library Preparation, Sequencing and Bioinformatic Analysis

Libraries, in duplicate, were prepared using 16S Barcoding kit (SQK-RAB204, Oxford Nanopore Technologies, Oxford, UK), according to the protocol. Amplicon purification was performed with AMPure XP Reagent (Beckman Coulter, Brea, CA, USA). The priming and loading were carried out with Cell Priming Kit (EXP-FLP002, Oxford Nanopore Technologies, Oxford, UK) according to the protocol, and the sequencing was performed on an R.9.4.1 MinION flow cell for 16 h. Reads were processed using the WIMP protocol on the EPI2ME platform (workflows 16S, v2023.04.21-1804452 and Antimicrobial Resistance v2023.04.26-1808834) [29]. All analyses were carried out in the laboratory of molecular biology and metagenomics at Stefan cel Mare University, Suceava, Romania.

### 2.4. Analysis of Diversity

The diversity and similarity through the samples were analyzed on the Galaxy Europe platform [30]. Alpha and beta diversity of bacteria species across samples was measured. To account for differences, the species with a number less than 100 reads were removed. We used 4 indices of alpha diversity to measure the diversity of the samples. Chao1 and abundance-based coverage estimator (ACE) were calculated to estimate the richness of the community from the samples. The dominance index and Sampson index were calculated to evaluate the dominance and the abundance of the species of each sample.

## 3. Results

Sample S4 had the highest values on the Chao1 and Simpson index, 214 and 0.98, respectively, and the lowest value on the dominance index, a value of 0.019. According to this indicator, sample S4 had high bacterial diversity, with the highest number of species but with low abundance. In contrast, sample S3 had the least species but in high abundance, with values of 101, 0.86, and 0.13 for the Chao1, Simpson, and dominance indices, respectively. The abundances were also high in the sample collected from the stream contaminated with effluents from the pork meat processing facility, S1 and S2, with values on the Simpson index of 0.92 and 0.91 and the dominance index of 0.79 and 0.83, respectively. The number of observed species is also relatively low, with a Chao1 index value of 106 and 108, respectively. The similarity through the samples at the species level was measured with beta diversity indices, Bray–Curtis, and binary Euclidean. The similarity is high in samples S1, S2, and S3, with many common species. The Bray–Curtis index had the highest value in samples S4 and S5, compared to other samples, which suggests that these samples are significantly different in terms of diversity, as also confirmed by the highest value for the Euclidean index, suggesting the dissimilarity of samples. The values of the Bray–Curtis index for sample S6 suggest a small degree of similarity with the first three samples.

The bacterial communities’ composition varied significantly among tested sites. Mainly, samples originating from Suceava river (sample from upstream and sample from the exit of the wastewater from the city’s sewage treatment plant) were clustered at order, family and genera level, while at the same levels, another cluster was formed by samples collected from the river that flows in front of the pork meat factory, from the Suceava river downstream from the discharge of the river, and from the stream that collects the wastewater from one neighborhood. The most abundant orders were *Burkholderiales* and *Pseudomonadales* for samples from upstream and from the exit of the city’s sewage treatment plant. The predominant family for these samples was *Burkholderiaceae* (Figure 3). The highest abundance was in sample S5, which was collected from the exit of the city’s sewage treatment plan and was the dominant family in samples S4, S5, and S6. Other abundant families were *Lentimicrobiaceae* in the sample from the exit of the treatment plant and Pseudomonadaceae in the sample upstream from the discharge of the river. The predominant genus for the two samples was *Rhodoferax* (Figure 4). The most abundant order for the cluster formed by the samples from the river that flows in front of the pork meat factory, from the Suceava river downstream from the discharge of the river and from the stream that collects the wastewater from one neighborhood, was *Campylobacterales*, being more abundant in the samples collected from the vicinity of the factory. At the family level, the most abundant was *Arcobacteraceae*. The predominant genus for these clustered samples was Arcobacter, with the highest abundance in the sample from the front of the pork meat factory.

### Antibiotic Resistance

Another line of analysis was the analysis of 16S antibiotic resistance genes. The samples were grouped (Figure 5) related to the place of sampling to observe differences between samples contaminated with meat factory residual waters (samples S1–S3) and samples containing only municipal waters (samples S4–S6). Based on the results, only ARGs which were above 0.2% of the total number of reads were further considered (Figure 6).

Of the 12 ARGs considered, only those for neomycin, streptomycin, and tobramycin were significantly higher in group G2, whereas tetracycline was significantly higher in G1. These differences can be correlated with the origin of the samples. According to the local wastewater collection plan [31], the samples in the second group originated from rivers/streams, which also collect residual water from two hospitals.

## 4. Discussion

In our research, *Arcobacteraceae*, a family found in high abundance in samples S1, S2, and S3, is widely distributed in various environments, including aquatic ones. The presence of species of this family in aquatic environments may be a threat to public health, as direct contact of humans or animals with some of these bacteria causes serious illness, with symptoms, such as diarrhea, abdominal pain, nausea, vomiting, and fever [32].

The high prevalence of the *Arcobacteraceae* family in S1, S2, and S3 stems from the number of reads identified as *Arcobacter suis CECT 7833*, a predominant bacteria isolated from pork meat [33], possibly related to the vicinity of the pork meat factory. The *Burkholderiaceae* family, which has a high abundance in all samples, is usually present in the root of the plants, and their abundance fluctuates with the presence of nitrate in the environment [34]. The residues from the pork meat factory are rich in nitrogen compounds, which decrease the abundance of *Burkholderiaceae* in samples S1, S2, and S3. Another abundant family in samples S1, S2, and S3 was *Moraxellaceae*, a family usually isolated from meat, which is recorded in high abundance, even in processed meat products [35].

At the genera level, a significant difference in abundance can be observed throughout the samples from the vicinity of the pork meat factory and the rest of the sample. The most abundant genera in samples S1, S2, and S3 are *Arcobacter*, *Aliarcobacter*, *Acidovorax*, and *Acinetobacter*. The most representative species of these genera were *Arcobacter suis CECT 7833*, *Acidovorax temperans*, *Acidovorax defluvii*, *Aliarcobacter cryaerophilus*, and *Acinetobacter johnsonii*. The high prevalence of *Arcobacter* is possibly due to the residues of the pork meat factory, bacteria from this genus being able to survive in unfavorable conditions, such as processing and the storage of meat [36]. Further, the high abundance of the *Aliarcobacter* genus reads is due to species such as *Aliarcobacter cryaerophilus* [37], which are potential foodborne pathogens. The *Aliarcobacter* species are members of intestinal microbiota of farm animals, being transmitted through feces and ending up contaminating the slaughterhouses and products [38]. *Acinetobacter* and *Acidovorax* are generally abundant in soils, wastewater, and wet environments, the latter being a pathogen for plants and specific to polluted environments [39,40]. In contrast to the bacteria diversity of samples S1, S2, and S3, samples S4, S5, and S6 have a high abundance of the genus *Rhodoferax*, their species being rich in freshwater environments [41], but some of these were also isolated from sewage, sludge, and sediments [42]. Another abundant genus in samples S4 and S6 is *Pseudomonas*, usually found in industrial equipment, aseptic solutions, cosmetics, medical products, and clinical instruments [43]. Their presence may be linked to the origin of the samples, these effluents collecting the wastewater from the two hospitals.

Antibiotics administered orally to patients are metabolized to a limited extent by the human body, being subsequently eliminated through excretion together with many antibiotic-resistant bacteria present in the patient’s intestine. Moreover, aminoglycosides, such as tobramycin, neomycin, and streptomycin, excreted through feces and urine in hospital wastewater, reach the environment. Here, they create a selective pressure on bacteria in favorable conditions, such as pH, temperature, and nutrients, and may lead to the emergence and spread of antibiotic resistance genes due to the horizontal gene transfer [44,45,46]. Neomycin, one of the antibiotics with many ARG reads in sample group G2, is usually administered with high efficiency against infection caused by Gram-positive (e.g., *Staphylococcus aureus*) or Gram-negative organisms, such as *Proteus vulgaris*, *Escherichia coli*, *Aerobacter aerogenes*, and *P. aeruginosa*. The presence of the neomycin ARGs in this group may be associated with the origin of samples, namely with the discharge of wastewater from hospitals in these places. Neomycin ARGs in our samples, identified as originating from several species, such as *Escherichia coli*, *Mycobacterium abscessus*, *Mycobacterium smegmatis*, and *Mycobacterium chelonae*, induce antibiotic target alterations through point mutations in the helix 44 region of the 16S rRNA [47]. In addition to neomycin, tobramycin is a common antibiotic found in hospital wastewater, and tobramycin ARGs may occur due to the presence of *Enterococcus* in such residual waters [48], as the aminoglycoside antibiotic exhibits poor penetration of the enterococcal cell envelope. The enterococcal metabolism is anaerobic, and the aminoglycosides transport, which is an oxygen-dependent process, cannot take place across the cytoplasmic membrane [49]. Tobramycin ARGs in the G2 sample group were identified as belonging especially to *Escherichia coli K-12* and *Mycobacterium abscessus*, and these genes also induce antibiotic target alterations through point mutations in the 3’ minor domain of the 16S rRNA gene [50]. Also, streptomycin ARGs belonging to *Escherichia coli K-12*, *Mycobacterium tuberculosis H37Rv*, and *Mycobacterium smegmatis str. MC2 155* were identified, occurring due to point mutations in the 5’ domain of the *rrsB* 16S rRNA gene.

The high value of tetracycline ARGs in sample group G1 may also be associated with the origin of these samples, which were collected from near a pork meat products factory. Tetracycline class antibiotics are used in swine farming, where they play different roles, such as growth promotors, for therapeutic purposes to treat diseases, and as prophylactic or metaphylactic treatment to prevent diseases [51]. The form of administration can be oral or via injection but is, most frequently, through inclusion of antibiotics in swine feeds. A large population of pigs receives medicated feed, thus being administered antimicrobial compounds with the ability to inhibit the growth of certain microorganisms [52]. Once ingested, antibiotics reach the meat of animals, where they favor the development of antibiotic resistance genes. The ARGs are present in all processing stages of pork meat. The antibiotics used in animal foods are structurally similar to antibiotics used in human health, and their utilization is a public health concern through the issue of resistant bacteria. The consumption of pork meat products is a transmission route of ARGs from animals to humans, because the ARGs are present in all processing steps of pork meat [53]. The tetracycline-resistant genes were identified as belonging to *Helicobacter pylori 26695*, *Escherichia coli K-12*, and *Propionibacterium acnes*. The mechanism of resistance assumes the binding of tetracycline tightly to the helix 34 domain in 16S rRNA, where it interferes sterically with the binding of aminoacyl-tRNA to the bacterial ribosomes, decreasing the binding affinity [54]. Antibiotic resistance in industrial processes can appear from multiple sources, such as the presence of the antimicrobials themselves (antibacterials, antifungals, and antiparasitic), heavy metals, and/or disinfectants and surfactants (biocides). Moreover, in addition of the emergence of resistance to a certain substance/molecule, there can also be a mechanism of co-resistance, where the selection of one gene can favor the selection of another gene, which, although it brings a selective advantage, is not directly related to the metabolic pathway of antimicrobial compounds. Also, cross-resistance can occur, where the same resistant gene can provide protection against several types of antimicrobial substances. Once they reach the environment, the half-life of antibiotics varies from a few hours to hundreds of days, but generally, antibiotics are considered molecules with high persistence in natural environments [53].

## 5. Conclusions

Following the nanopore sequencing of municipal waters and wastewaters from various sampling sites, the rapid assessment of diversity of bacterial communities becomes feasible. Our results showed that water microbiota composition is subject to various influences, especially from human industrial activities, which may lead to contamination with potential pathogens or to horizontal gene transfer. Our data showed that microbial diversity can drop by 50% in waters contaminated with effluents from meat processing facilities, as the Chao1 diversity index reduces from >200 to around 100. In parallel, the Simpson diversity index reduces up to 12% in the same contaminated waters compared to uncontaminated, flowing river water. The higher abundance of *Pseudomonas* genera, naturally found in waters, is replaced by the abundance of potential enteropathogens or zoonotic agents such as *Arcobacter*. Also, as seen with the data we obtained, the vicinity of biotechnological units such as meat processing factories can induce local environments in municipal waters where antibiotic resistance can develop or spread, considering the fact that, for instance, tetracycline resistance genes occur with up to 50% more incidence in contaminated waters. Moreover, specific antibiotic resistance may occur, which is of epidemiological but also research interest.

Such incidence of biological processes is not routinely assessed but offers molecular data that can be used to find better solutions for protecting the environment and human health. The accurate, genomic characterization of microbial diversity is valuable, especially if performed in real time or mobile setups and can serve as tools for epidemiologists but also for urban planners or industrial process optimization with respect to environmental values. Mobile, nanopore technology represents a useful tool for achieving such goals; moreover, when it is coupled with accessible, fast bioinformatics, a potential avenue is the development of mobile units for the molecular-based microbiological assessment of municipal and residual waters.

While this study reveals the composition and diversity of microbial communities in municipal wastewaters using an affordable and mobile sequencing technology, there are some limitations that must be taken into account: first, the experimental design should consider seasonal variability and other factors (grazing or other anthropogenic activities next to the water) that may lead to unaccounted, at the moment, specificities of the microbial communities’ composition; second, the nucleic acid extraction can be optimized, so as to make sure that all microbial taxonomic categories are represented accurately; third, although the accessibility of the Epi2me platform greatly improves speed and automation, the data obtained should be run through different bioinformatic workflows so as to determine the most accurate approach in taxonomical classification.

## Figures and Tables

**Figure 1 microorganisms-11-02834-f001:**
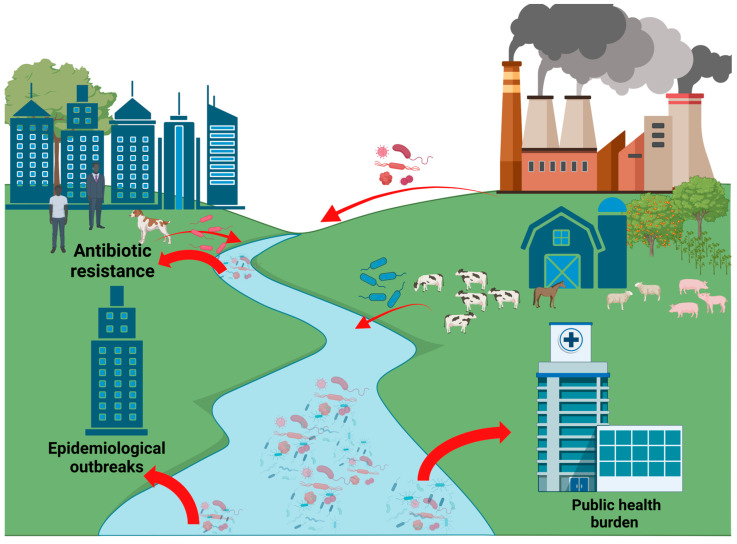
The sources of bacterial pathogens in the environment (red arrows indicate potential microbial spread).

**Figure 2 microorganisms-11-02834-f002:**
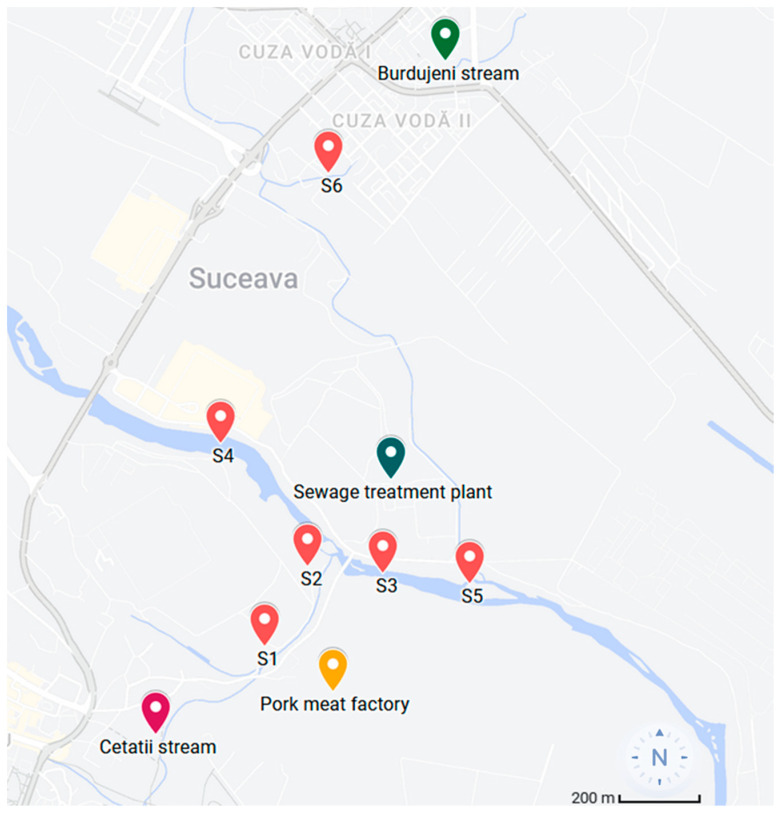
Wastewater sample collection sites.

**Figure 3 microorganisms-11-02834-f003:**
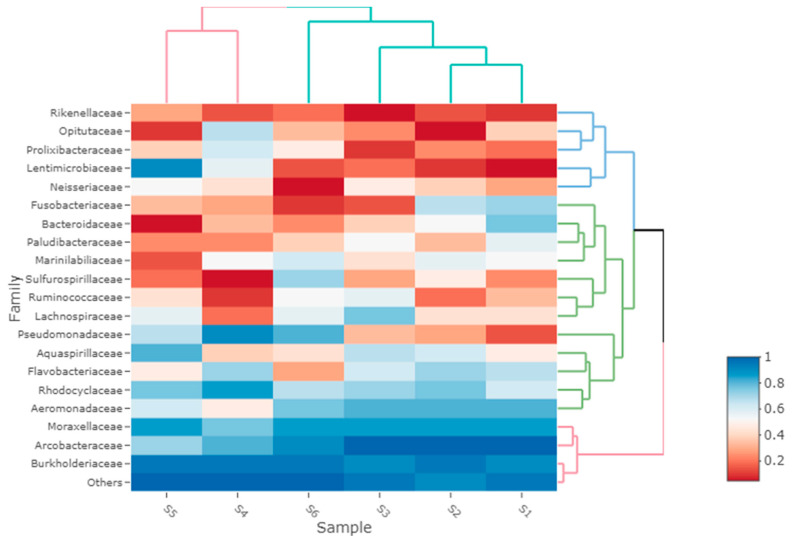
Predominant bacterial families for the six environmental samples.

**Figure 4 microorganisms-11-02834-f004:**
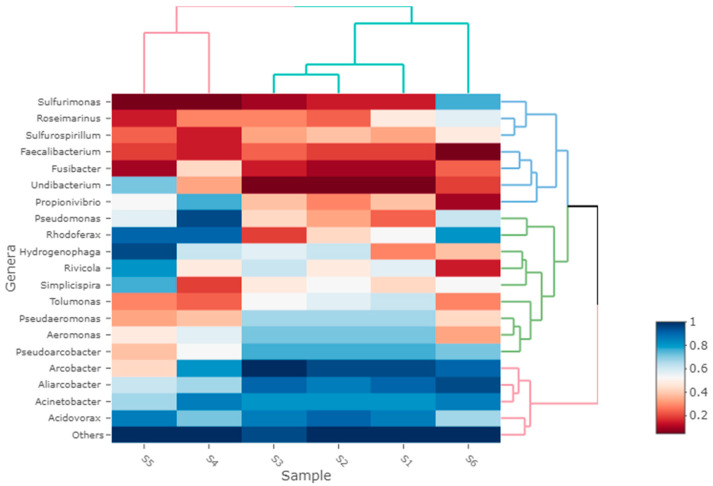
Predominant genera in the six environmental samples.

**Figure 5 microorganisms-11-02834-f005:**
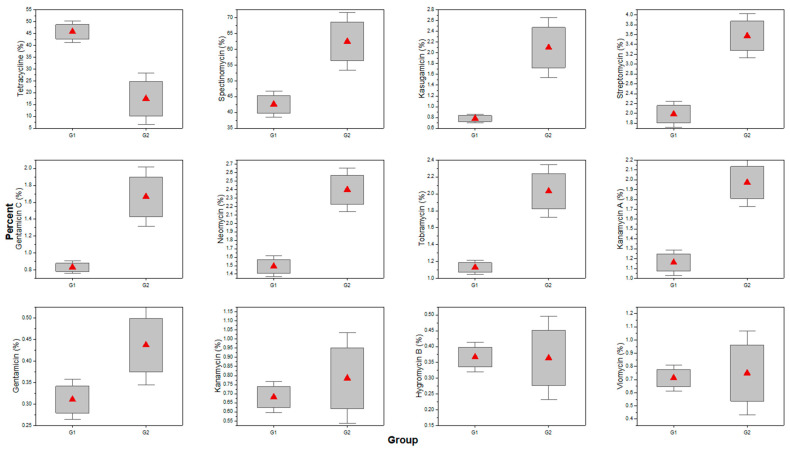
Incidence (% total 16S rRNA reads) of antibiotic resistance genes between the two groups of samples (G1—samples contaminated with meat factory residual waters, G2—samples containing only municipal waters; red triangle indicates mean values).

**Figure 6 microorganisms-11-02834-f006:**
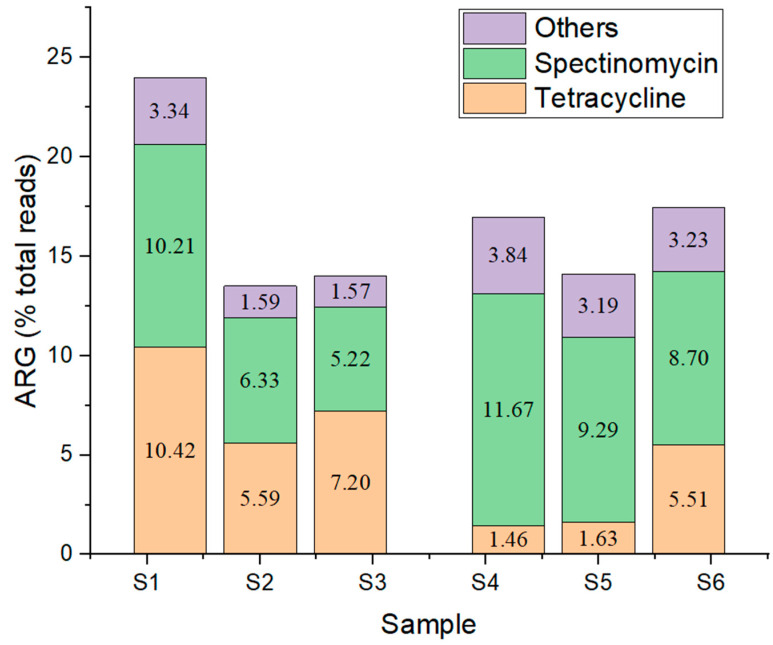
Distribution (% total reads) of main antibiotic resistance genes within samples (ARG—antibiotic resistance genes).

## Data Availability

Data are contained within the article.

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
