# Peer review of "Nanopore Sequencing Assessment of Bacterial Pathogens and Associated Antibiotic Resistance Genes in Environmental Samples"

_microorganisms, 2023, doi:10.3390/microorganisms11122834_

Round 1

Reviewer 1 Report

Comments and Suggestions for Authors

The manuscript by Andrei Lobiuc et al. investigated the bacterial pathogens and associated antibiotic resistance genes in environmental samples. The results are interesting. I have the following questions and comments:

1, for the sampling process, how many replicates were used? The authors must provided relevant information in the manuscript. 

2, the y axis of figure 6 needs to be specified. 

3, The limitations of the study must be discussed.

4, The new discoveries of the present study must be clearly presented. This is not a new topic. 

Author Response

The suggestions were welcomed and addressed in their entirety, thank you!

Reviewer 2 Report

Comments and Suggestions for Authors

The framework of this manuscript has promised results and the manuscript is organized and well-written, however some changes are required. Please find my comments below:

*The abstract is weak, you need to include the main results/findings of your research along with the implications. The Abstract section should be comprehensive and should stand alone. From where samples were taken? Romania?

*Keywords: add another 4 keywords that reflect the content of your study.

*Add reference for the following statement “with usual numbers for bacteria such as E. coli, ranging from 2.5x103 to 4.4x105 colony forming units (CFU)/100 ml or total coliforms up to 7.9x105 CFU/100 ml.”

*The introduction section needs improvement. You have to mention previous research that have successfully detected the bacterial pathogens. You need also to mention previous research that used Next Generation Sequencing (NGS) techniques.

*Refer to Figure 1 in the text.

*Add a new Table that contains the coordinates of the sampling points.

*Move Figure 2 caption under the figure.

*You need to show the north direction and the scale bar in map of Figure 2

*Section 2.2: add reference for the applied protocol

*line 147: It is not appropriate to add link within the text “(https://epi2me.nanoporetech.com/)”. Move it to the reference list. Do the same for line 151 “(https://usegalaxy.eu)”

*When these samples were collected? and what is the name of the lab in which the samples were analyzed? is it analyzed in triplicate?

*In the conclusion section: You need to add the main results/findings of your research along with the implications.

*Please add the limitations of your study at the end of the conclusion section.

Comments on the Quality of English Language

Moderate editing of English language required.

Author Response

(The authors gave the same response as above.)

Round 2

Reviewer 1 Report

Comments and Suggestions for Authors

The authors have revised the manuscript. It can be considered for publication.

Author Response

Thank you for your constructive suggestions.

Reviewer 2 Report

Comments and Suggestions for Authors

Thanks for addressing most of my comments, however, I still need to see more improvement in both abstract and introduction section. In the abstract, your main findings still not presented. In the introduction section, you need to mention previous research that have successfully detected the bacterial pathogens along with their results, it supposed to be the basis of your research. Your research is a complementary to literature research, not stand alone research. Also, the conclusions section is still not  supported by the results/numbers.

Comments on the Quality of English Language

 Minor editing of English language required.

Author Response

Thank you!
